# Association of Hepatobiliary Phase of Gadoxetic-Acid-Enhanced MRI Imaging with Immune Microenvironment and Response to Atezolizumab Plus Bevacizumab Treatment

**DOI:** 10.3390/cancers15174234

**Published:** 2023-08-24

**Authors:** Yosuke Tamura, Atsushi Ono, Hikaru Nakahara, Clair Nelson Hayes, Yasutoshi Fujii, Peiyi Zhang, Masami Yamauchi, Shinsuke Uchikawa, Yuji Teraoka, Takuro Uchida, Hatsue Fujino, Takashi Nakahara, Eisuke Murakami, Masataka Tsuge, Masahiro Serikawa, Daiki Miki, Tomokazu Kawaoka, Wataru Okamoto, Michio Imamura, Yuko Nakamura, Kazuo Awai, Tsuyoshi Kobayashi, Hideki Ohdan, Masashi Fujita, Hidewaki Nakagawa, Kazuaki Chayama, Hiroshi Aikata, Shiro Oka

**Affiliations:** 1Department of Gastroenterology, Graduate School of Biomedical and Health Sciences, Hiroshima University, Hiroshima 734-8551, Japan; 2Cancer Treatment Center, Hiroshima University Hospital, Hiroshima 734-8551, Japan; 3Department of Diagnostic Radiology, Graduate School of Biomedical and Health Sciences, Hiroshima University, Hiroshima 734-8551, Japan; 4Department of Gastroenterological and Transplant Surgery, Graduate School of Biomedical and Health Sciences, Hiroshima University, Hiroshima 734-8551, Japan; 5Laboratory for Cancer Genomics, RIKEN Center for Integrative Medical Sciences, Yokohama 230-0045, Japan; 6Collaborative Research Laboratory of Medical Innovation, Graduate School of Biomedical and Health Sciences, Hiroshima University, Hiroshima 734-8551, Japan; 7Research Center for Hepatology and Gastroenterology, Hiroshima University, Hiroshima 734-8551, Japan; 8Department of Gastroenterology, Hiroshima Prefectural Hospital, Hiroshima 734-8530, Japan

**Keywords:** hepatocellular carcinoma, Gd-EOB-DTPA-enhanced MRI, β-catenin mutation, surrogate marker, immune microenvironment, atezolizumab plus bevacizumab therapy

## Abstract

**Simple Summary:**

High intensity of gadolinium ethoxybenzyl diethylenetriamine pentaacetic acid (Gd-EOB-DTPA)-enhanced MRI imaging (EOB-MRI) in the hepatobiliary phase (HB) is associated with mutations in *CTNNB1* and activation of β-catenin, an immune-cold microenvironment, and an unfavorable response to anti-PD-1/PD-L1 monotherapy in patients with hepatocellular carcinoma (HCC). EOB-MRI could serve as a surrogate marker predicting the immune microenvironment and molecular subtype but does not predict the response to atezolizumab + bevacizumab therapy. Our results suggest that this is because the high-intensity group benefits from bevacizumab, while the low-intensity group benefits from atezolizumab. Although EOB-MRI might serve as a surrogate marker for the response to other currently developed immunotherapies, it is not necessary to avoid atezolizumab + bevacizumab treatment for hyperintense HCC.

**Abstract:**

It has been reported that high intensity in the hepatobiliary (HB) phase of Gd-EOB-DTPA-enhanced MRI (EOB-MRI) is associated with an immune-cold microenvironment in HCC. The aim of this study is to reveal whether non-high-intensity HCCs are homogeneous with respect to the immune microenvironment and to investigate the predictive ability of EOB-MRI for the response to atezolizumab + bevacizumab therapy (Atezo/Bev). The association between differences in stepwise signal intensity of HB phase and molecular subtypes and somatic mutations associated with the immune microenvironment was investigated in 65 HCC patients (cohort 1). The association between EOB-MRI and the therapeutic effect of Atezo/Bev was evaluated in the Atezo/Bev cohort (60 patients in cohort 2). The proportion of HCCs having *CTNNB1* mutations and classified as Chiang CTNNB1 and Hoshida S3 was high in the high-intensity HB-phase group. Infiltration of tumor-associated macrophages (TAM) and regulatory T-lymphocytes (Treg) was characteristic of the high-intensity and low-intensity groups, respectively. Although EOB-MRI could not predict the response to Atezo/Bev treatment, our results demonstrate that EOB-MRI could serve as a surrogate marker predicting the immune microenvironment. This suggests that Atezo/Bev treatment can be selected regardless of signal intensity in the EOB-MRI HB phase.

## 1. Introduction

Several molecular classifications of hepatocellular carcinoma (HCC) have been established [1,2]. In the era of immune checkpoint inhibitors (ICI), molecular classes representing different immune microenvironments have received attention. HCC can be classified into two major molecular groups: the proliferation class and the non-proliferation class [3,4,5,6,7]. The proliferation class is characterized by more aggressive tumors with poor histological differentiation [8]. The proliferation class can be further divided into two subclasses: Hoshida S1 or Boyault G2–G3, characterized by Wnt–TGFβ activation; and Hoshida S2 or Boyault G1, characterized by a progenitor-like phenotype with the expression of stem cell markers (CK19 and EPCAM) and a high level of AFP expression [4,7]. Hoshida S1/Boyault G2–G3 tumors are thought to be characteristic of immune-exhausted microenvironments with high immune infiltration; therefore, they are more likely to respond to ICI therapy [9].

The non-proliferation class is characterized by less-aggressive tumors with better histological differentiation [8]. The non-proliferation class can be further divided into two subclasses: the Wnt/β-catenin CTNNB1 subclass presents frequent *CTNNB1* mutations and activation of the Wnt/β-catenin signaling pathway; and the interferon subclass is characterized by activation of the IL6–JAK–STAT signaling pathway and a more inflamed tumor microenvironment [2]. The Wnt/β-catenin CTNNB1 subclass represents an immune-excluded phenotype with low immune infiltration [3,7,10]; therefore, patients with this subclass may respond less well to ICI therapy [9].

It is known that enhanced uptake of gadoxetic-acid-enhanced magnetic resonance imaging (EOB-MRI) in the hepatobiliary (HB) phase is associated with mutations in *CTNNB1* [9] and activation of β-catenin [11]. Therefore, it is expected that EOB-MRI could serve as a surrogate marker for the tumor immune microenvironment.

However, only approximately 10% of HCCs are classified as high-intense in the HB phase, and we considered that the non-high-intense tumors, which represent the majority of HCCs, are unlikely to be homogeneous. Therefore, one of the purposes of this study was to assess whether the degree of the low intensity of the HB phase could play a role in predicting the molecular subclass affecting the immune microenvironment among non-high-intense tumors.

In solid tumors, combinations of ICI and other drugs have been reported to be more effective than ICI alone [12]. Actually, combination treatment with the anti-programmed death-ligand 1 (PD-L1) monoclonal antibody atezolizumab plus the anti-vascular endothelial growth factor (VEGF) agent bevacizumab (Atezo/Bev) demonstrated prognostic superiority over sorafenib therapy in a phase III trial and was globally approved in 2020 as a primary systemic chemotherapy for unresectable HCC [13].

Biomarkers that predict the response to ICI therapy in HCC patients include PD-L1 expression, tumor mutational burden (TMB), microsatellite instability (MSI) status, and gut microbiota [14,15]. It has been reported that the intensity of the nodule in the HB phase of EOB-MRI is a promising imaging biomarker for predicting an unfavorable response with anti-PD-1/PD-L1 monotherapy in patients with HCC [16]. However, little is known about whether EOB-MRI could predict the response to Atezo/Bev treatment. A secondary purpose of the study was to investigate whether the intensity of the EOB-MRI in the HB phase could predict the response to Atezo/Bev treatment.

In some studies, the uptake was evaluated only in the HB phase [17,18]; however, no standardized method has been established for EOB-MRI [19,20]. Aside from the evaluation method, previous studies have commonly shown that tumors appearing iso-high in the HB phase frequently harbor *CTNNB1* mutations.

## 2. Materials and Methods

### 2.1. Study Design and Patients

In this study, we examined the following hypotheses: (1) If there is a linear correlation between the degree of decreased uptake and decreased expression of OATP1B, the characteristics of hepatocellular carcinoma related to tumor immunity may not change nominally with increased/decreased uptake but may change stepwise with the degree of decreased uptake; and (2) if the above hypothesis is confirmed, then the degree of uptake of EOB-MRI may predict the response to ICI. To test hypothesis (1), we investigated the relationship between MRI signal intensity and *OATP1B1* or *OATP1B3* expression in 65 HCC cases for which RNA-Seq and preoperative EOB-MRI had previously been performed (cohort 1). To test hypothesis (2), we investigated the relationship between EOB-MRI intensity and treatment effect in 60 HCC patients treated with Atezo/Bev (cohort 2) (Figure 1). Sixty-five HCC patients who met the following conditions were enrolled in cohort 1: (1) underwent hepatectomy at Hiroshima University Hospital between 2009 and 2012, (2) underwent pre-operative Gd-EOB-DTPA-enhanced MRI imaging before surgery, (3) and underwent whole genome sequencing (WGS) and RNA-Seq for tumors in a previous study as part of the International Cancer Genome Consortium (ICGC) LIRI-JP project [21]. Mutation calling from whole-genome sequencing was performed by the Pan-Cancer Analysis of Whole Genomes (PCAWG) project [22]. All patients provided written informed consent for their participation in the study, which followed the ICGC guidelines and those of the institutional review boards at RIKEN and Hiroshima University. Cohort 2 consisted of 60 unresectable HCC patients who started Atezo/Bev between October 2020 and November 2022 at Hiroshima University Hospital and who underwent EOB-MRI imaging prior to treatment. The Human Ethics Review Committee of Hiroshima University approved the study (E-624-5). All patients provided written informed consent. Patient characteristics of cohorts 1 and 2 are shown in Table 1 and Table 2, respectively.

### 2.2. RNA-Seq and WGS

RNA-Seq data were obtained from the ICGC data portal (https://dcc.icgc.org/ (accessed on 18 August 2021)). As the ICGC data matrix had been normalized in FPKM, it was converted to TPM using the following formula: TPM=(FPKM∑FPKM)×106.

Genes were grouped with respect to key HCC pathways according to a previous report [23] (Appendix A). We defined cases that had a mutation in any of the genes included in those pathways as being positive for pathway mutations. The mutations annotated with the following gene ontology terms were extracted using the ICGC portal: missense, frameshift, splice acceptor, splice donor, splice region, stop gained, and disruptive inframe deletion.

### 2.3. Molecular Subclasses of HCC

Five gene sets based on Chiang’s classification [3] and three gene sets for Hoshida’s classification [4] were downloaded from MSigDB (v6.2). For Fujita’s immunological classification [24], we used the following four markers: fraction of (i) M2 macrophages (tumor-associated macrophages (TAMs)) and (ii) regulatory T cells (Tregs) estimated by CIBERSORT, (iii) Wnt/β-catenin signaling signature computed by a single-sample gene set enrichment analysis (GSEA) with gene lists acquired from the literature [24], and (iv) cytolytic activity (CYT), which is defined as the average expression of granzyme A (GZMA) and perforin (PRF1) [25]. The subclasses were determined by the nearest template prediction method [26] using GenePattern v3.9.11 (https://www.genepattern.org/ (accessed on 27 September 2021)).

In our previous study, in which we established the Fujita classification [24], we performed immunohistochemical staining of FOXP3 and CD163 and confirmed that they are positively correlated with the CIBERSORT scores for Treg and TAM, respectively.

### 2.4. Gene Set Enrichment Analysis

GSEA software v4.1.0 was downloaded from the GSEA website (https://www.gsea-msigdb.org/gsea/index.jsp (accessed on 11 March 2020)). GSEA was performed as described previously [27] to analyze the differential modulation of molecular pathways in the Hallmark gene sets (h.all.v7.4), using the following parameters: permutation type = gene_set, scoring scheme = weighted, metric for ranking genes = Signal2Noise, max gene set size = 500, min gene set size = 3.

### 2.5. MRI Interpretation

The signal intensities (SIs) of the tumor and surrounding background liver were measured by defining regions of interest (ROIs) following the method used in previous reports [20,28]. Tumor ROIs were determined by tracing the margin of the tumor that would be considered the largest, even though the signal intensity (SI) was heterogeneous in each of the ROIs devoid of necrosis. ROIs on the adjacent liver parenchyma were determined by tracing the surrounding nontumorous region within approximately 20 mm from the tumor while avoiding vascular structures. The definitions and formulas are as follows: SInod and SIpar refer to the SI of the nodule and liver parenchyma, respectively. The relative intensity ratio (RIR) = SInod/SIpar, and the relative enhancement ratio (RER) equals RIRpost/RIRpre, where RIRpost is the RIR in the HB phase images, and RIRpre is the pre-contrast RIR. In the current study, we defined the top and bottom quartiles of each ratio as the high and low groups, respectively. Ratios in between were assigned to the intermediate (int) group. Typical correspondences of tumor images to RIR and RER are shown in Appendix A. We classified patients into three groups according to RIRpost and RER as follows: high (top 25%), intermediate (middle 50%), and low (bottom 25%) in cohort 1. The images were evaluated by two hepatologists who were blinded to the treatment effect. RIRpost and RER values and groups in cohort 1 are shown in Appendix A.

### 2.6. Statistical Analysis

Statistical analysis was performed using JMP Pro 14.0.0 (SAS Institute Inc., Cary, NC, USA). Intergroup differences were tested using the Mann–Whitney U test or the Fisher’s exact test for continuous or categorical variables, respectively. The Cochran–Armitage trend test was used to assess the presence of an association between a variable with two categories and an ordinal variable with k categories. Inter-correlations of the gene expression levels of *OATP1B1* and *OATP1B3* and the genes constituting the CHIANG_LIVER_CANCER_SUBCLASS_CTNNB1_UP, consisting of genes up-regulated in the CTNNB1 subclass, and CHIANG_LIVER_CANCER_SUBCLASS_CTNNB1_DN, consisting of genes down-regulated in the CTNNB1 subclass, were assessed by Pearson’s correlation coefficient. The progression-free survival (PFS) of RECIST during the Atezo/Bev treatment was estimated using the Kaplan–Meier method, and differences among subgroups were evaluated using the log-rank test.

EOB-MRI was performed at one time point before the start of treatment, and tumor size and contrast effects were evaluated by contrast-enhanced CT or EOB-MRI before and a median of every 1–2 months during treatment. In Atezo/Bev treatment, the second efficacy was determined by contrast-enhanced CT or EOB-MRI approximately 3–4 months after the start of treatment.

For continuous variables, the median value was used as a threshold if no specific cutoff had been established. All comparisons were considered significant at a *p*-value less than 0.05.

## 3. Results

### 3.1. Relationship between Molecular Class and Expression Levels of OATP1B1 and OATP1B3

It is known that Gd-EOB-DTPA undergoes specific organic anion transporting protein (OATP1B and OATP1B3)-dependent hepatocyte uptake at the canalicular membrane of hepatocytes [29,30]. Therefore, we investigated the relationship between the molecular class and the RNA expression levels of *OATP1B1* and *OATP1B3*, which are associated with EOB uptake. The expression level of OATP1B and EOB-MRI uptake were positively correlated (Figure 2a).

It was found that the genes in the CHIANG_LIVER_CANCER_SUBCLASS_CTNNB1_UP set had a positive correlation with expression of both *OATP1B1* and *OATP1B3* (Figure 2b). On the other hand, genes in the CHIANG_LIVER_CANCER_SUBCLASS_CTNNB1_DN set showed a negative correlation. Both *OATP1B1* and *OATP1B3* were upregulated in Chiang CTNNB1 and Fujita WNT classes, while *OATP1B1* was also upregulated in Hoshida S3 (Figure 2c,d).

### 3.2. Clinical Characteristics of HCC Patients with Respect to RIRpost/RER Degree

Preoperative serum DCP levels were higher in the RIRpost-low group than in the RIRpost-int group (*p* = 0.0498). Prothrombin time (PT) % was lower in the RIRpost-high group than in the RIRpost-int group or the RIRpost-low group (*p* = 0.046 or 0.0476, respectively). The RER-high group consisted of a higher proportion of HCV patients and a lower proportion of HBV patients compared to the RER-low group. The proportion of poorly differentiated tumors was lower in the RER-high group compared to the RER-int group. The RER-int group showed a higher neutrophil lymphocyte ratio (NLR) and a higher platelet lymphocyte ratio (PLR) than the RER-low and RER-high group, respectively (Table 3).

### 3.3. Association of Mutation Status and RIRpost/RER

There were no significant differences in RIRpost with respect to the mutation status of *CTNNB1* or in the Wnt/β-catenin pathway. On the other hand, RER was significantly higher in patients with a mutation in *CTNNB1* or the Wnt/β-catenin pathway than in those without (*p* = 0.0010 and 0.0363, respectively) (Figure 3a). Genes were grouped with respect to key HCC pathways according to a previous report [23] (Appendix A). The mutations annotated with the following gene ontology terms were extracted using the ICGC portal: missense, frameshift, splice acceptor, splice donor, splice region, stop gained, and disruptive inframe deletion. Interestingly, the distribution of patients with mutations in *CTNNB1* or in the Wnt/β-catenin pathway increased in a stepwise fashion from the low to high group in RER (*p* = 0.0024 and 0.0156, respectively) (Figure 3c) but not in RIRpost (*p* = 0.2027 and 0.3027, respectively) (Figure 3b). The frequencies of patients with somatic mutations in frequently mutated genes and pathways are shown in Appendix A.

### 3.4. Association of Molecular Classes and RIRpost/RER

RIR was significantly lower in Chiang’s proliferation class compared to that in the CTNNB1 or polysomy 7 class (*p* = 0.0063 and 0.0058, respectively) and higher in Hoshida S3 compared to the S2 class (*p* = 0.0208). The distribution of the Hoshida S3 class was significantly higher in the RIR-high than the combination of the -intermediate and -low groups (*p* = 0.0003) (Figure 4a). On the other hand, there was no difference between inter-subclasses in either Chiang or Hoshida’s classification for RER (Figure 4b).

Figure 4e,f,h,i summarize the association of RIRpost/RER and Chiang and Hoshida subclasses, respectively. For Chiang’s subclass, the frequency of patients with the CTNNB1 subclass was significantly higher in the RIRpost-high group than in the RIRpost-intermediate/low group (*p* = 0.0208). Interestingly, the proportion of patients in the proliferation subclass increased in a stepwise fashion from the high to the low groups in RIRpost (*p* = 0.0250, respectively). On the other hand, the proportion of patients in the CTNNB1 subclass decreased from the high to the low groups in RER (*p* = 0.043).

### 3.5. Association of Immune Microenvironment and RIRpost/RER

We previously reported that primary liver cancer can be classified into four subclasses using the following four markers representing the immune microenvironment: TAM, Treg, Wnt/β-catenin signaling, and CYT score [24]. The WNT signaling score was significantly higher in the RIRpost-high group than in the RIRpost-intermediate or RIRpost-low group (*p* = 0.0219 and 0.0044, respectively) (Appendix A). TAM score was significantly higher in the RIRpost-high group than in the RIRpost-intermediate or RIRpost-low group (*p* = 0.0321 and 0.0332, respectively) (Figure 4c). Treg score was significantly higher in the RIRpost-low group than in the RIRpost-high group (*p* = 0.0287) (Figure 4d). There was no significant difference in CYT score with respect to classification with either RIRpost or RER (Appendix A). Fujita’s TAM class decreased in a stepwise fashion from the RIRpost-high to RIRpost-low groups (*p* = 0.0482) (Figure 4g). The opposite tendency was observed in the Treg class, although there was no statistical significance using the Cochran–Armitage trend test. The proportion of each subclass was not significantly different between RER high/int/low (Figure 4j).

### 3.6. Angiogenesis Was Enhanced in the RIRpost-High Group

GSEA analysis revealed that the HALLMARK_ANGIOGENESIS gene set was enriched in RIRpost-high compared to RIRpost-low (FDR q value 0.0287, NES 1.47) and RIRpost-int than RIRpost-low (FDR q value 0.0560, NES 1.35) (Figure 5a). The gene expression level of *VEGFR2* was significantly higher in RIRpost-high than RIRpost-low (*p* = 0.0079) (Figure 5b). All other GSEA data are shown in Appendix A.

### 3.7. EOB-MRI Imaging Was Not Predictive of Atezo/Bev Treatment Benefit

We investigated whether pre-treatment EOB-MRI imaging could predict the response in Atezo/Bev-treated patients in an independent cohort (cohort 2). To evaluate the relationship between EOB-MRI imaging and the response to treatment, the relationship between EOB-MRI imaging and treatment initiation and tumor growth was investigated using conventional RECIST criteria.

The Kaplan–Meier curve showed no significant difference in PFS between the RIRpost and RER-high, -int, and -low groups (Figure 6a,b). There was also no significant difference in PFS when divided by cutoff of 0.9 for RER (Appendix A).

We investigated the relationship between RIRpost/RER and the rate of growth of the main tumor during the second efficacy test of Atezo/Bev treatment (Figure 6c). There were 26 cases in which the size of the main tumor increased, with a maximum increase of 113%, and 34 cases in which the size of the main tumor decreased, with a maximum decrease of 67%.

A tumor with a high RIR (1.23) showed no enlargement (Figure 6d), while a tumor with a low RIR (0.49) showed enlargement in patient AB9 (Figure 6e).

### 3.8. Comparison between Two Tumors in Cohort 2

In cohort 2 (Atezo/Bev treatment group), efficacy was determined by RECIST, and RIRpost/RER was compared for each tumor if there were target lesions other than the main tumor. However, 37 cases in which the target lesion was extrahepatic or had lymph node metastasis or in which the tumor could not be recognized by plain MRI were excluded (Appendix A). The results showed no significant difference in RIRpost/RER between the two tumors. The rate of tumor growth was generally consistent, but there were cases, such as case 3 and case 40, in which one tumor grew, while the other remained unchanged.

## 4. Discussion

We classified RIRpost and RER into three groups., namely high (top 25%), intermediate (middle 50%), and low (bottom 25%), and examined their association with *CTNNB1* mutations and molecular class. The frequency of *CTNNB1* and Wnt/β-catenin mutations increased significantly in RER from the low to high groups (Figure 3a).

The proportion of Hoshida S3 or Chiang CTNNB1 class was significantly higher in the RIRpost-high group than in the other groups. In contrast, the proportion of Chiang’s proliferation class increased from the RIRpost-high to -low groups in a step-by-step manner (Figure 4e,f). With respect to the immune microenvironment markers used for Fujita’s classification, TAM and WNT scores were higher in the RIRpost-high group, while the Treg score was higher in the RIRpost-low group. To summarize, RER is superior for predicting *CTNNB1* mutation but inferior to RIRpost for predicting molecular subclass.

Our results show that high EOB-MRI intensity is correlated with CTNNB1 subclass (Chiang/Hoshida S3) as well as the Fujita TAM class, but this is in conflict with a previous report. For example, the M2 macrophage signature is more aggregated in the immune-exhausted class, which is a subclass of the immune class reported by Sia et al. [10]. However, in our previous report, the TAM and CTNNB1 subclasses represented non-inflamed tumors with lower CYT compared to other subclasses. Likewise, only 13% and 2% of the TAM and CTNNB1 subclasses were in Sia’s immune class, whereas 64% and 51% of the CYT and Treg subclasses were present in Sia’s immune class, respectively [24]. One possible reason for this is that the signatures did not represent a trade-off relationship. There were cases with a concomitance of high cytolytic activity and high M2 signature and of high M2 signature and WNT signaling signature. Based on that, the Fujita subclass was determined based on which subclass was the relatively strongest. Another possibility is that differences in the methods by which macrophages or M2 macrophages were defined could account for this discrepancy. However, in our previous study, we confirmed that CIBERSORT estimates of Tregs and TAM were positively correlated with the immunohistochemistry of FOXP3 and CD163, respectively. It might be said that immune-cold HCC can be divided into subclasses characterized by TAM, WNT, and Treg [24].

It is well known that TAM contributes to the immune-suppressive phenotype [31,32]. It has also been reported that activation of Wnt/β-catenin in TAM and HCC is associated [33]. In other words, high intensity HCC in the EOB-MRI HB phase is consistent with TAM. Our findings suggest that TAM might play an important role in the mechanism underlying the poor response to the immune checkpoint inhibitor in high RIRpost/RER HCC.

We then investigated the relationship between the angiogenesis pathway, expression of *VEGFA* and its receptors *VEGFR1* and *VEGFR2*, and EOB-MRI imaging and revealed that the angiogenesis pathway was more activated in the higher RIRpost groups. The gene expression level of *VEGFR2*, a receptor of *VEGFA*, was significantly higher in the RIRpost-high group than in the RIRpost-low group. VEGFR2 is considered to be the major mediator of the mitogenic, angiogenic, and permeability-enhancing effects of VEGF [34]. Bevacizumab, a monoclonal antibody against VEGF-A, is thought to primarily target VEGFA-VEGFR2 signaling, and we hypothesized that bevacizumab would be more effective if VEGFR2 was expressed in HCC. Accordingly, we hypothesized that anti-angiogenic treatments such as bevacizumab may provide more benefit in the case of high RIRpost/RER HCC.

Finally, the imaging and immune microenvironment of HCC is said to be heterogeneous even among tumors from the same patient. Such tumor-to-tumor heterogeneity may lead to therapeutic resistance, concern for which has recently gained importance. In this study, we were most interested in the relationship between image quality and Atezo/Bev treatment response. The intensity of the HB phase was not predictive of PFS during Atezo/Bev treatment. We hypothesized that high RIRpost/RER HCC is unlikely to benefit from ICI monotherapy but is likely to benefit from anti-angiogenic treatment. In other words, these results suggest that Atezo/Bev treatment can be selected, and a therapeutic effect can be obtained regardless of high or low RIRpost/RER HCC in the EOB-MRI HB phase.

Meanwhile, dual immune checkpoint blockade with durvalumab plus tremelimumab has recently been approved as an effective first-line treatment for unresectable HCC based on the results of the HIMALAYA phase III trial [35]. If the above hypothesis is correct, EOB-MRI might be useful for predicting the response to dual ICI therapy, as the therapy does not contain an anti-angiogenic agent.

A comparison of background factors between cohort 1 and cohort 2 showed that age was significantly higher in cohort 2, but there was no difference in sex. In terms of tumor factors, the cohort 2 group was significantly more advanced in T-factor, stage, and differentiation, but there was no difference in tumor size. As for liver function, albumin was significantly higher in cohort 1, and Child–Pugh class was higher in the cohort 2 group (Appendix A). The difference in PFS in the cohort 2 group may not be significant, as the signal intensity in the HB phase of EOB-MRI is known to be affected by liver function.

In addition, the comparison between the two tumors in cohort 2 showed no significant differences in RIRpost or RER, but there were differences in tumor growth in some cases (Appendix A); thus, combining other factors in addition to MRI findings may be a more accurate predictor of HCC treatment response.

## 5. Conclusions

We revealed that in addition to the difference between high- and non-high RIRpost/RER classification, the degree of RIRpost/RER was also useful to stratify molecular subtypes among HCC tumors. Although the intensity of the HB phase in EOB-MRI was not able to predict the response to Atezo/Bev treatment, it might serve as a surrogate marker for the response to other therapies currently in development.

### Limitations

This study has several important limitations. First of all, the sample size is small. Most importantly, the cut-off values of RIR and RER were determined based on the distribution in the cohort; however, the imaging conditions were not fully consistent since patients who had MRIs taken at different facilities and at different ages were included. It would have been ideal to examine the effect of immunotherapy in the cases in which we saw an association between EOB-MRI and immune molecular subtypes, but we decided to prepare a separate cohort because only two of the RNA-Seq cases had been treated with immunotherapy for postoperative recurrence. We were not able to analyze VEGFR in cohort 2 due to the lack of availability of pathology samples.

## Figures and Tables

**Figure 1 cancers-15-04234-f001:**
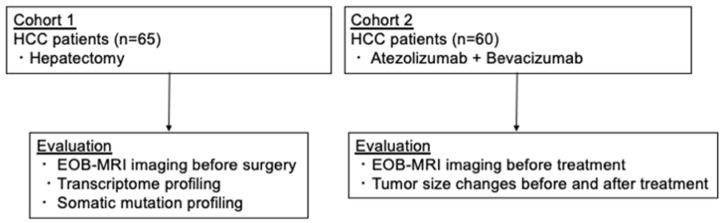
Flow diagram of the objective HCC patients.

**Figure 2 cancers-15-04234-f002:**
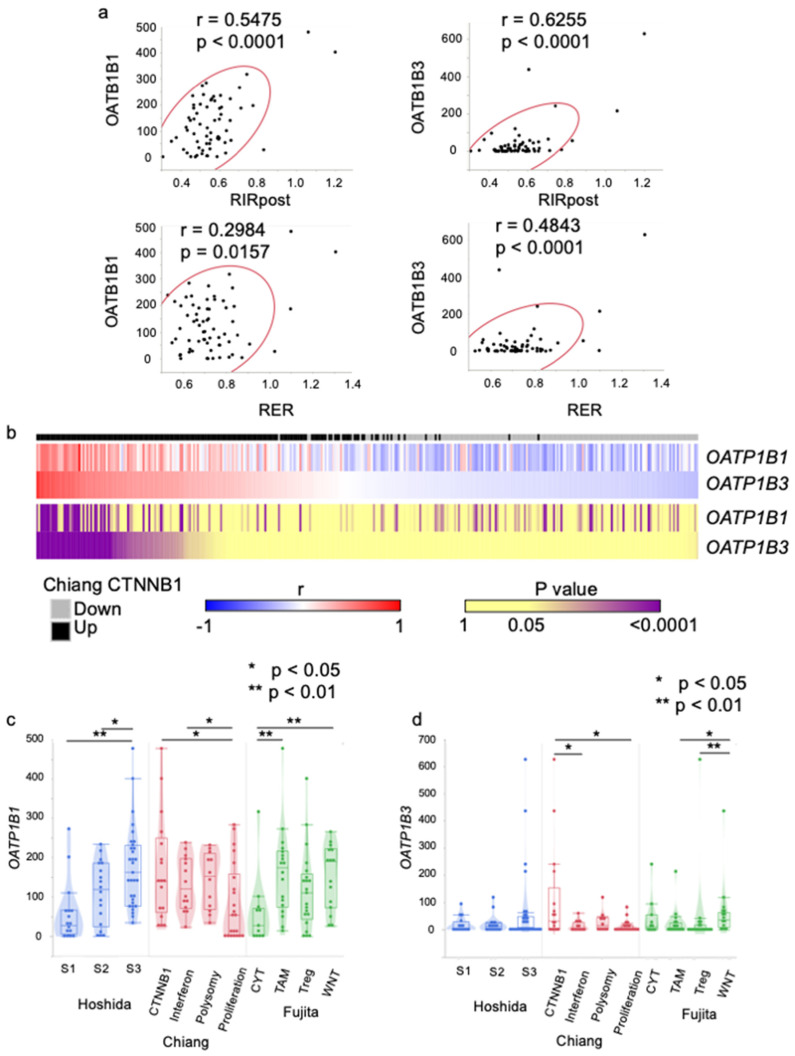
Association of the gene expression levels of *OATP1B1*/*OATP1B3* and *CTNNB1* pathway and molecular subclass. (**a**) Scatter plots showing the correlation between (upper left) OATP1B1 and RIRpost, (upper right) OATP1B3 and RIRpost, (lower left) OATP1B1 and RER, (lower right) and OATP1B3 and RER. (**b**) A heat map summarizing the inter-correlations of the gene expression levels of *OATP1B1*/*OATP1B3* and the genes constituting the CHIANG_LIVER_CANCER_SUBCLASS_CTNNB1. Each column represents a gene belonging to the CHIANG_LIVER_CANCER_SUBCLASS_CTNNB1_UP (black) or CHIANG_LIVER_CANCER_SUBCLASS_CTNNB1_DN (gray). Blue-red and yellow-purple colors represent the ρ and FDR q values assessed by Pearson’s correlation coefficient. Gene expression levels of *OATP1B1* (**c**) and *OATP1B3* (**d**) according to the molecular subclass. (* *p* < 0.05, ** *p* < 0.01).

**Figure 3 cancers-15-04234-f003:**
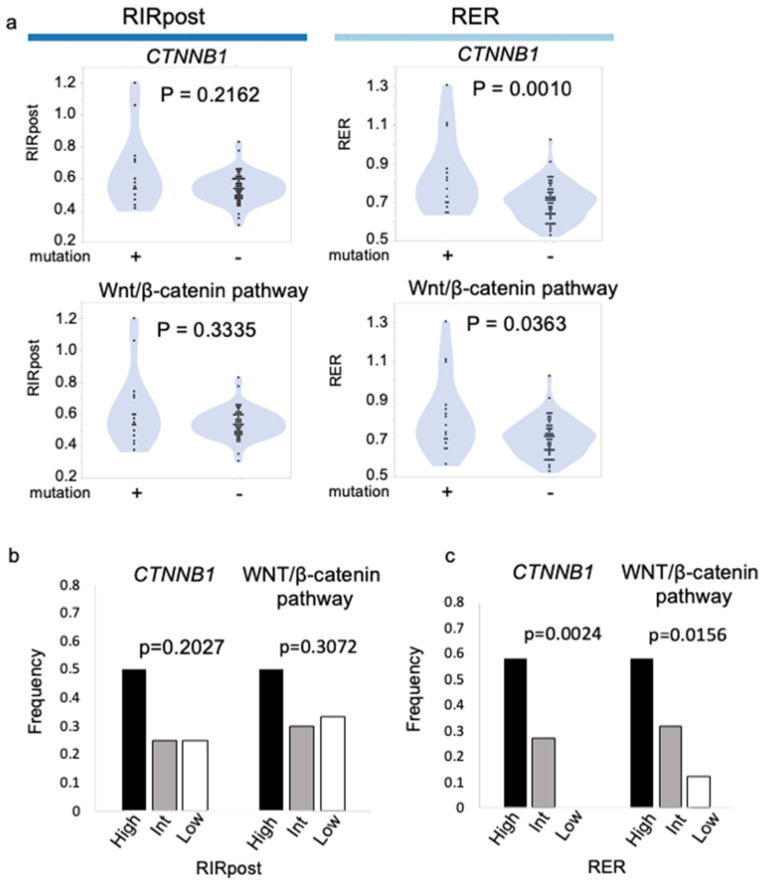
Association of mutation status and RIR/RER. (**a**) RIRpost according to the mutation status of *CTNNB1* (upper left) and Wnt/β-catenin pathway (*CTNNB1, AXIN1,* and *APC*) (lower left). RER according to the mutation status of *CTNNB1* mutation (upper right) and Wnt/β-catenin pathway (lower right). The frequencies of patients with a mutation in *CTNNB1* and the Wnt/β-catenin pathway according to RIRpost (**b**) and RER (**c**) high/intermediate/low groups.

**Figure 4 cancers-15-04234-f004:**
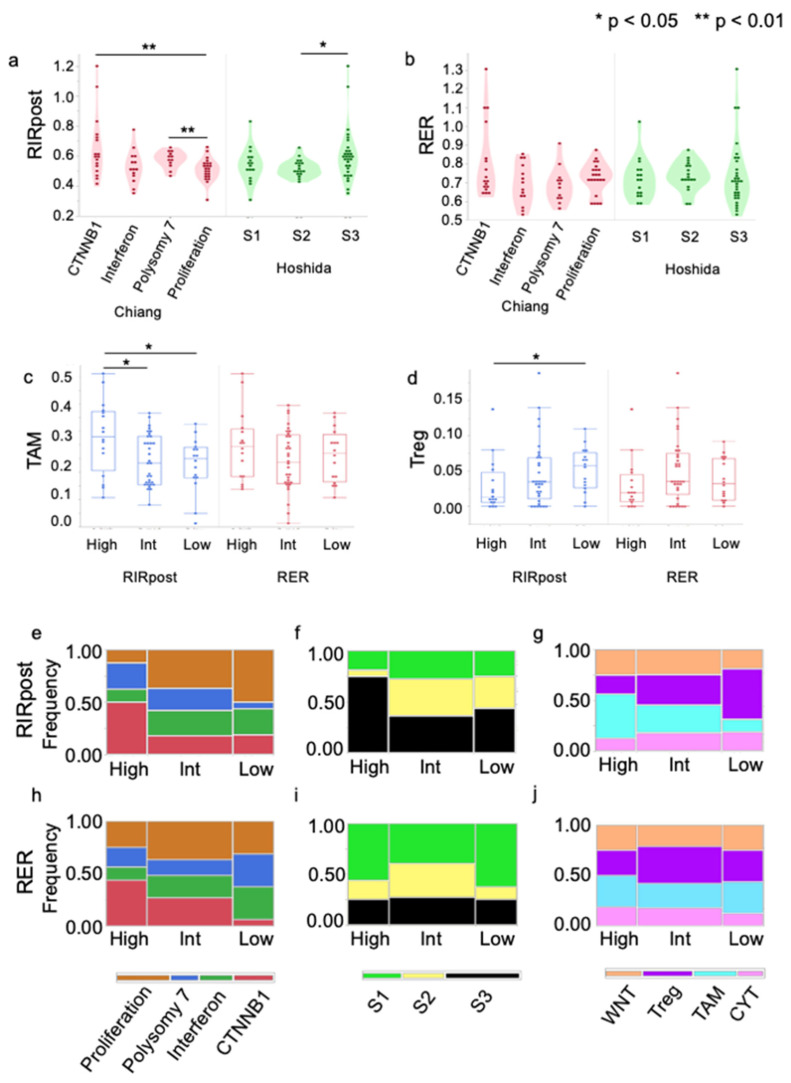
Association of molecular classes and RIR/RER. RIRpost (**a**) and RER (**b**) according to the Chiang and Hoshida subclass. * *p* < 0.05; ** *p* < 0.01. TAM (**c**) and Treg (**d**) score according to the RIRpost and RER classification. (**e**) Mosaic plots showing the distribution of the Chiang, (**f**) Hoshida, and (**g**) Fujita subclass with respect to RIRpost-high/intermediate/low groups. (**h**) Mosaic plots showing the distribution of the Chiang, (**i**) Hoshida, and (**j**) Fujita subclass with respect to RER-high/intermediate/low groups.

**Figure 5 cancers-15-04234-f005:**
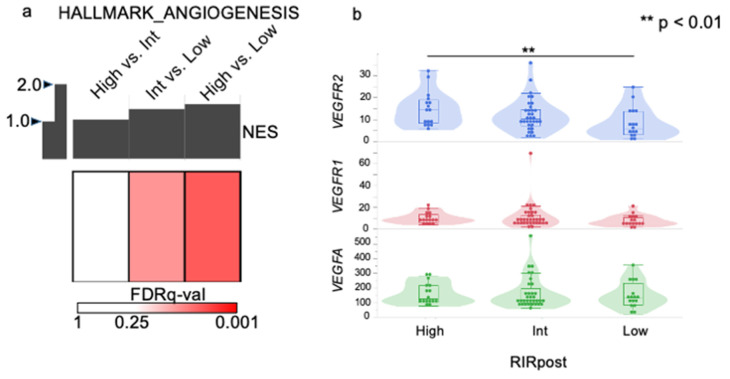
Angiogenesis was enhanced in RIRpost-high tumor. (**a**) Summarized GSEA results for the HALLMARK_ANGIOGENESIS gene set. The top bar and white-red color represent the NES and FDR q-value assessed by GSEA to investigate the difference between the RIRpost-high and -low group, -int and -low, and -high and -int group. NES was enriched in the former group, showing positive values. GSEA, gene set enrichment analysis; NES, normalized enrichment score; FDR, false-discovery rate. (**b**) The gene expression levels of *VEGFA*, *VEGFR1,* and *VEGFR2* according to the RIRpost-high, -int, and -low group.

**Figure 6 cancers-15-04234-f006:**
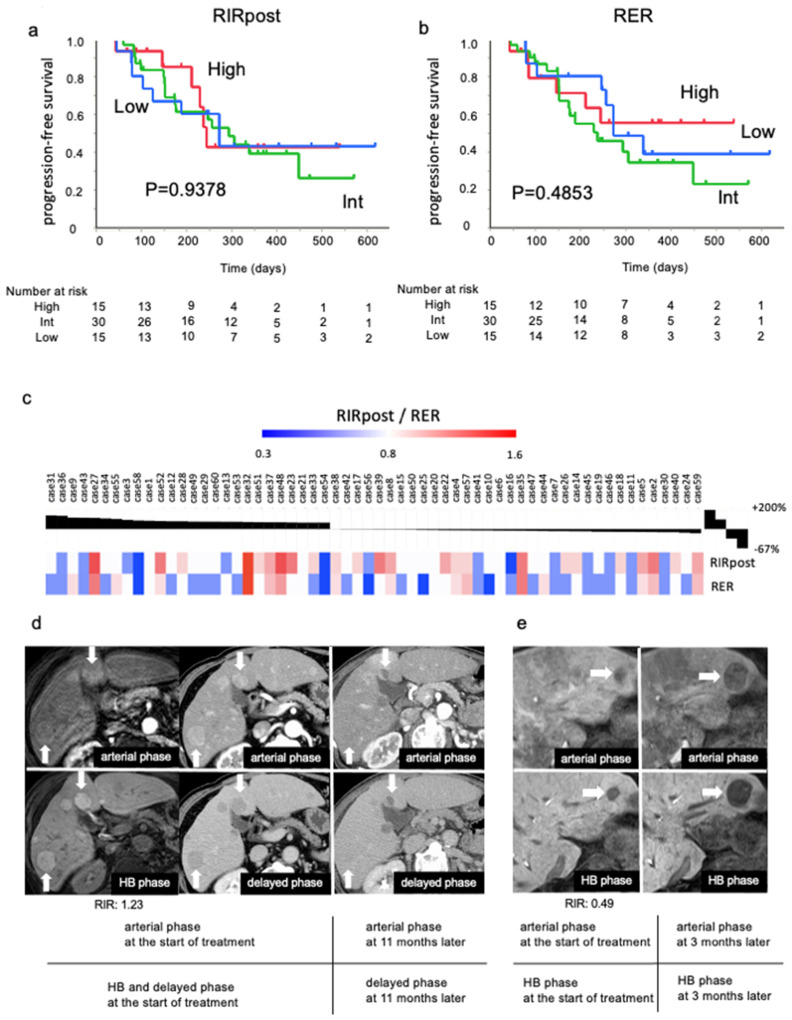
Evaluation of PFS between EOB-MRI image intensities in Atezo/Bev treatment. A Kaplan–Meier curve representing progression-free survival (PFS) of the tumor according to RIRpost (**a**)/RER (**b**). (**c**) A waterfall plot showing RIRpost/RER of the main tumors and the rate of tumor growth at the second efficacy determination. MRI or CT images of HCC patients receiving Atezo/Bev treatment. (**d**) High-intensity HCCs with respect to the response to Atezo/Bev treatment. (**e**) Low-intensity HCCs with respect to tumor enlargement.

**Table 1 cancers-15-04234-t001:** Patient characteristics of cohort 1.

Variable	*n* = 65
Age (years)	67 (31–89)
Sex (female/male)	13/52
Etiology (HBV/HCV/NBNC)	18/32/15
T (1/2/3/4)	15/26/20/4
Main tumor size (mm)	25 (12–160)
Treatment before surgery (TACE/none)	29/36
Differentiation (well/mod/poor)	9/48/8
AFP (ng/mL)	7.1 (1–43,700)
DCP (AU/L)	146 (5–58,889)
White blood cells (/mm^3^)	5120 (1820–9440)
Neutrophils (/mm^3^)	3393 (1180–6898)
Lymphocytes (/mm^3^)	1478 (329–3112)
Platelets (×10^4^/mm^3^)	14.5 (4–47.1)
NLR	2.40 (0.75–6.67)
PLR	104.1 (32.3–262.5)
PT (%)	90 (33–128)
Albumin (g/dL)	4.3 (3.1–5.1)
Total bilirubin (mg/dL)	0.7 (0.4–1.7)
AST (IU/L)	30 (17–100)
ALT (IU/L)	29 (11–175)
γGTP (IU/L)	52 (19–552)
Warfarin +/−	4/61
Child–Pugh class (A/B/C)	65/0/0
Types of MRI	1.5 T

AFP, alpha-fetoprotein; DCP, des-gamma-carboxy pro-thrombin; AST, aspartate aminotransferase; ALT, alanine aminotransferase; γGTP, γ-glutamyl transpeptidase; HBV, hepatitis B virus; HCV, hepatitis C virus; NBNC, non-B non-C; NLR, neutrophil-lymphocyte ratio; PLR, platelet–lymphocyte ratio.

**Table 2 cancers-15-04234-t002:** Patient characteristics of cohort 2.

Variable	*n* = 60
Age (years)	72 (49–92)
Sex (female/male)	11/49
Etiology (HBV/HCV/NBNC)	8/19/33
T (1/2/3/4)	0/16/39/5
Main tumor size (mm)	34 (10–130)
N (+/−)	5/55
M (+/−)	7/53
HCC stage (2/3/4a/4b)	11/34/8/7
Differentiation (well/mod/poor/ND)	31/21/3/5
AFP (ng/mL)	7.1 (1.0–9689)
DCP (AU/L)	277.5 (11–35,040)
White blood cells (/mm^3^)	5285 (1670–11,620)
Neutrophils (/mm^3^)	2945 (860–9760)
Lymphocytes (/mm^3^)	1185 (420–2790)
Platelets (×10^4^/mm^3^)	14.3 (3.2–42.2)
NLR	2.35 (0.70–11.8)
PLR	107.2 (31.4–324.6)
PT (%)	92 (64–124)
Albumin (g/dL)	3.9 (2.5–4.6)
Total bilirubin (mg/dL)	0.8 (0.3–2.0)
AST (IU/L)	32 (14–80)
ALT (IU/L)	28 (7–136)
Child–Pugh class (A/B/C)	54/5/1
Types of MRI	3 T

AFP, alpha-fetoprotein; DCP, des-gamma-carboxy pro-thrombin; AST, aspartate aminotransferase; ALT, alanine aminotransferase; HBV, hepatitis B virus; HCV, hepatitis C virus; NBNC, non-B non-C; NLR, neutrophil-lymphocyte ratio; PLR, platelet–lymphocyte ratio; ND, no data.

**Table 3 cancers-15-04234-t003:** Clinical characteristics of HCC patients with respect to the degree of RIRpost/RER.

	RIRpost						RER					
	High (0.61–1.20)	Int (0.48–0.60)	Low (0.31–0.48)	*p*			High (0.79–1.31)	Int (0.64–0.79)	Low (0.53–0.64)	*p*		
				High vs. Int	High vs. Low	Int vs. Low				High vs. Int	High vs. Low	Int vs. Low
Sex (female/male)	1/15	10/23	2/14	0.0762	1	0.2898	3/13	9/24	1/15	0.7261	0.5996	0.1347
Age	67 (57–81)	69 (31–89)	67 (32–86)	0.7086	0.4836	0.6772	68.5 (57–89)	67 (31–84)	67.5 (47–78)	0.3472	0.2272	0.6929
Etiology (HBV/HCV/NBNC)	3/9/4	9/17/7	6/6/4	0.8476	0.4418	0.6735	2/10/4	8/18/7	8/4/4	0.7049	0.049	0.1156
Main tumor size (mm)	22.5 (12–58)	25 (12–160)	31.5 (13–150)	0.2157	0.1624	0.7327	25 (12–150)	28 (12–150)	26 (14–160)	0.9065	0.8502	0.8896
T (1/2/3/4)	4/9/3/0	8/14/9/2	3/3/8/2	0.807	0.0567	0.2295	3/9/4/0	9/10/11/3	3/7/5/1	0.3351	0.8631	0.8631
Differentiation (well/mod/poor)	4/11/1	4/23/6	1/14/1	0.3742	0.4809	0.5114	4/12/0	2/25/6	3/11/2	0.0448	0.5252	0.4224
AFP (ng/mL)	5.8 (2–7168)	17.5 (1–29,180)	16.5 (3.8–43,700)	0.1111	0.3528	0.9914	12.9 (2–7168)	7.1 (1–43,700)	5.35 (2.1–3749)	0.6991	0.6358	0.3222
DCP (AU/L)	39.5 (5–13,641)	242 (5–48,328)	228 (18–58,889)	0.0498	0.0704	0.7898	50 (5–31,481)	178 (5–58,889)	123.5 (6.7–5972)	0.3267	0.5977	0.7572
White blood cells (/mm^3^)	4650 (2700–9000)	5370 (1820–9100)	5020 (3070–9440)	0.5434	0.4623	0.7981	4650 (2700–9000)	5680 (1820–9100)	4635 (2700–9440)	0.2243	0.7919	0.3215
Neutrophils (/mm^3^)	2505.5 (1220–6345)	3421 (1180–6898)	3499 (2002–6232)	0.5434	0.2351	0.4428	2505.5 (1250–6345)	3806 (1220–6212)	2856.5 (1220–6212)	0.1442	0.8358	0.1594
Lymphocytes (/mm^3^)	1155.5 (721–3112)	1521 (329–2248)	1285 (690–2647)	0.0985	0.5591	0.3763	1371.5 (721–2187)	1478 (329–2248)	1584 (690–3112)	0.6933	0.6109	0.1898
NLR	2.24 (1.14–6.11)	2.40 (0.75–5.08)	2.61 (1.51–6.67)	0.8898	0.1809	0.1171	2.12 (1.14–3.19)	2.58 (0.98–6.67)	1.83 (0.75–5.14)	0.0683	0.5847	0.0232
Platelet count (×10^4^/mm^3^)	12.85 (4.9–47.1)	15.3 (4–38.4)	13.25 (7.4–23.3)	0.5155	0.4176	0.8898	13.95 (4.9–18.8)	14.6 (4–33.9)	14.25 (5.8–47.1)	0.4685	0.3664	0.8562
PLR	96.6 (48.6–213.4)	100 (32.3–262.5)	122.4 (49.1–208.9)	0.9745	0.2662	0.1075	81.6 (48.6–213.4)	112.3 (49.1–216.9)	107.0 (32.3–262.5)	0.0396	0.2206	0.5434
PT (%)	83 (33–109)	90 (39–128)	94.5 (74–112)	0.046	0.0476	0.693	89 (33–109)	92 (42–128)	91.5 (74–112)	0.267	0.4279	0.8645
Albumin (g/dL)	4.3 (3.4–4.7)	4.3 (3.1–5.1)	4.4 (3.8–4.8)	0.4872	0.1782	0.3921	4.1 (3.1–5.0)	4.4 (3.3–5.1)	4.25 (3.2–4.7)	0.1483	0.2726	0.5927
Total bilirubin (mg/dL)	0.75 (0.4–1.6)	0.7 (0.4–1.4)	0.7 (0.5–1.7)	0.4556	0.9092	0.7612	0.7 (0.4–1.4)	0.7 (0.4–1.7)	0.75 (0.4–1.4)	0.9135	1	0.914
AST (IU/L)	37.5 (17–100)	29 (17–82)	29.5 (18–55)	0.579	0.2823	0.7488	41 (18–100)	28 (17–82)	30 (19–70)	0.0878	0.3086	0.5149
ALT (IU/L)	31 (13–77)	31 (12–175)	25 (11–57)	0.932	0.1414	0.0899	34.5 (13–77)	29 (11–114)	25.5 (12–175)	0.4362	0.8358	0.7168
γGTP (IU/L)	56.5 (17–552)	51 (12–304)	49.5 (9–99)	0.9745	0.7628	0.5576	61 (17–552)	51 (9–256)	45.5 (12–304)	0.4177	0.4738	0.932

AFP, alpha-fetoprotein; DCP, des-gamma-carboxy pro-thrombin; AST, aspartate aminotransferase; ALT, alanine aminotransferase; γGTP, γ-glutamyl transpeptidase; HBV, hepatitis B virus; HCV, hepatitis C virus; NBNC, non-B non-C; NLR, neutrophil–lymphocyte ratio; PLR, platelet–lymphocyte ratio; PT, prothrombin time.

## Data Availability

The original contributions presented in the study are included in the article/Appendix A. Further inquiries can be directed to the corresponding authors.

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
