# Peer review of "Association of Hepatobiliary Phase of Gadoxetic-Acid-Enhanced MRI Imaging with Immune Microenvironment and Response to Atezolizumab Plus Bevacizumab Treatment"

_cancers, 2023, doi:10.3390/cancers15174234_

Round 1

Reviewer 1 Report (Previous Reviewer 1)

The manuscript has been significantly improved according to the reviewers' suggestions.

Author Response

Thank you for your peer review.

Reviewer 2 Report (Previous Reviewer 2)

Authors have added the required data and comments. Overall, I find this to be a good manuscript.

I only have a few minor concerns after reading this revised draft, as follows;

Discussion and Figure.6 (Line.394 and 405)

What is the meaning/definition of “second efficacy”?  Please explain this in the method paragraph.

Figure.6 (Line.402)
Please change the sentence. For example, “evaluation of PFS between,,,,”

Conclusion (Line. 503)

What is the meaning of “non RIR post/RER HCC tumors”? Please modify these words.

Limitation

Please add that the sample size is small.

Author Response

Discussion and Figure.6 (Line.394 and 405)

What is the meaning/definition of “second efficacy”?  Please explain this in the method paragraph.

 Response: We apologize for the lack of explanation. We have added the following sentence to the methods paragraph (Line.253-255):

In Atezo/Bev treatment, the second efficacy was determined by contrast-enhanced CT or EOB-MRI approximately 3-4 months after the start of treatment.

Figure.6 (Line.402)
Please change the sentence. For example, “evaluation of PFS between,,,,”

Response: Thank you for your suggestion. We have changed the text in Fig. 6 as follows: Evaluation of PFS between EOB-MRI image intensities in Atezo/Bev treatment 

Conclusion (Line. 503)

What is the meaning of “non RIR post/RER HCC tumors”? Please modify these words.

Response: We apologize for the confusion. We have modified "non-RIR post/RER HCC tumors" to "HCC tumors".

Limitation 

Please add that the sample size is small.

Response: Thank you for your comment. We have added to the Limitation (Line.511).

This manuscript is a resubmission of an earlier submission. The following is a list of the peer review reports and author responses from that submission.

Round 1

Reviewer 1 Report

The authors investigated the association of signal intensity of tumors on the hepatobiliary phase of EOB-MRI with immune microenvironment. In addition, they investigated whether signal intensity of tumors could predict the response to Atezolizumab plus Bevacizumab therapy. The concept of this study is interesting; however, there are several concerns.

1) Cohort 1 (HCC patients treated with hepatectomy) and cohort 2 (HCC patients treated with Atezolizumab plus Bevacizumab) appear to have different patient characteristics, including liver function and tumor factors. Signal intensity of liver parenchyma on the hepatobiliary phase of EOB-MRI has been reported to be influenced by liver function. Therefore, it is necessary to show the similarities and differences between these two cohorts.

2) It is not clear how to interpret the signal intensity when patients have multiple HCC. How the authors assess the difference of signal intensity among tumors in such situation?

3) Please explain the methods and results of Fig 6a in more detail.

Reviewer 2 Report

Comment:

Although the study population is small, it is very interesting concept to estimate the tumor immune microenvironment using EOB-MRI. However, the author's classification of tumor signal intensity is based not on visual assessment or RER, which are common methods of evaluating tumor signal intensity of EOB-MRI in previous papers, but on RIR. This might mislead the readers about hyperintense HCC. In fact, Figure 2 a shows only a small number of HCCs with an RIR of 1 or higher. Author should avoid the expression of “hyperintense HCC” in main text. Author also needs to further clarify the reasons and significance of the RIR.

Abstract Line.46

It seems difficult to conclude the efficacy of EOB-MRI in predicting the response due to the small population.

Abstract Lines.96.-97

This is not to say that no evaluation method has been established. In my opinion, there is no standardized evaluation method.

Materials and Methods (2.1.) Figure 1

Modify the number of patients in Cohort 2 in Figure 1 to match the number of patients in the main text.

Materials and Methods (2.1.) Table 1 and 2

Authors should add the types of MRI scanner in table 1 and 2 (1.5T or 3.0T)

Also should add Child-Pugh class in table 1.

Materials and Methods (2.5.) Line.160

In this study, RIR is used for the evaluation of tumor signal intensity. RER has been used to quantitatively assess tumor signal intensity since previous reports have showed that RER (optimal cut off=0.9) is significantly associated with Wnt/b-catenin-activated HCC, as cited by the author in the main text. Although author wrote why RIR was used (in Lines 384-391), author should explain in more detail why RIR is more useful in the evaluation of tumor signal intensity than RER. In my opinion, it should be carefully evaluated to determine Wnt/b-catenin-activated HCC using RIR, since RIR is highly influenced by the background liver signal intensity and pre-contrast tumor signal intensity.

Materials and Methods (2.7.) Line.176

Author should describe the ROC curve for the optimal cut off of RIR.

Results (3.2.) Line.221

Author should show the range of RIR in each groups.

Results (3.5.) Line.263

Although the concept is interesting that TAM score is significantly higher in RIR-high group, author need to clarify some points. At first, how many high RER groups (optimal cut off=0.9) are included in the high RIR group? A table of the correlation between RER (high/low) and RIR(high/int/low) in this study would be very helpful.

Author also should write why high-RIR HCC is associated with TAM based on the correlation between imaging and molecular aspect. In the article(No.33) author cited in the main text, nuclear accumulation of b-catenin is seen in the M2-like macrophage, not HCC cell. Based on this article, I am wondering HCC with M2-TAM show low-RIR HCC since HCC cell does not express OATP1B3.

At last, I feel RIR might not be specific enough to determine two different subclass due to the limitation mentioned above. Author, at least, should show the previous reports using RIR to demonstrate the usefulness of evaluating tumor signal intensity.

Results (3.7.) Line.293

Figure 6 figure and description do not match.

Discussion (4) / Conclusion(5)

In my opinion, the phrase of “high-intensity HCC” is not appropriate in this article.